# Look-Ahead Selective Plasticity for Continual Learning of Visual Tasks

**Rouzbeh Meshkinnejad**
Department of Computer Science
Western University
London, ON N6A3K7
{rmeshkin}@uwo.ca

**Jie Mei** *
Department of Computer Science
Western University
London, ON N6A3K7
{jie.mei}@it-u.at

**Zeduo Zhang**
Department of Computer Science
Western University
London, ON N6A3K7
{zzhan762}@uwo.ca

**Daniel Lizotte**
Department of Computer Science
Western University
London, ON N6A3K7, Canada
{dlizotte}@uwo.ca

**Yalda Mohsenzadeh**
Department of Computer Science
Western University
London, ON N6A3K7, Canada
{ymohsenz}@uwo.ca

## Abstract

Contrastive representation learning has emerged as a promising technique for continual learning as it can learn representations that are robust to catastrophic forgetting and generalize well to unseen future tasks. Previous work in continual learning has addressed forgetting by using previous task data and trained models. Inspired by event models created and updated in the brain, we propose a new mechanism that takes place at task boundaries, i.e., when one task ends and another begins. By observing the redundancy-inducing ability of contrastive loss on the output of a neural network, our method uses the first few samples of the new task to identify and retain parameters that contribute most to the transferability of the neural network, freeing up the remaining parts of the network to learn new features. We evaluate the proposed methods on benchmark computer vision datasets including CIFAR10 and TinyImagenet, and demonstrate state-of-the-art performance in task-incremental, class-incremental, and domain-incremental continual learning scenarios. [2]

## 1 Introduction

Deep neural networks (DNNs) have solved a variety of computer vision tasks with high performance. While this feat has been achieved by accessing large and diverse datasets, in many practical scenarios, the data is not initially available in its entirety and becomes available over time, potentially containing new unseen classes and different target distributions. When presented with a sequence of classification

---

[*] Also affiliated with Department of Anatomy, University of Quebec in Trois-Rivieres and Interdisciplinary Transformation University (ITU) Austria.

[2] The source code is available here.

Preprint.

tasks to learn and remember, DNNs suffer from a well-known catastrophic forgetting problem [1], abruptly losing their performance on previous classification datasets. To address this problem, various continual learning algorithms have been proposed.

Regularization-based methods aim to preserve knowledge from previous tasks by applying penalties to changes in weights or functions. Techniques like Elastic Weight Consolidation (EWC) [2], Synaptic Intelligence (SI) [3], RWalk [4], XKFAC [5], and one similar to our work, Attention-based Structural Plasticity [6] penalize parameter changes based on their importance to past tasks but often fail to consider the relevance of these parameters for future tasks. Other approaches, such as ResCL [7], AFEC [8], and [9], mitigate intransigence by renormalizing new task solutions with the old model. Techniques like Learning without Forgetting (LwF) [10], LwM [11], FROMP [12], and DRI [13] use knowledge distillation to apply regularization to intermediate or final outputs of the prediction function. Methods like NPC [14] focus on network dynamics, slowing down the learning rate for important neurons. However, despite these efforts, regularization-based methods still struggle with forward transfer, scalability, and inter-task interference in complex scenarios.

Rehearsal-based methods such as iCaRL [15], GSS [16], GEM [17], and CLS-ER [18], among others [19–23] store a small number of training samples from previous tasks in a memory, while other methods such as [24–29] train a generative model to produce training samples similar to previous tasks. During the training of the current task, the network is simultaneously trained on the current task samples as well as samples from the memory. A key challenge in these methods is selecting a subset of past samples that best represent all previous tasks. These subsets are often chosen through random sampling [30, 31], by proximity to class means [15], or using more advanced, gradient-based optimization techniques [19, 32]. In this work, we will use simple rehearsal and replay of samples as part of our proposed method to mitigate forgetting.

While methods that use shared parameters may lead to inter-task interference, architecture-based approaches address this by assigning separate network components through parameter isolation [33, 34] and dynamic architectures [35–37] for different tasks. Parameter isolation methods assign a distinct subset of the network's parameters to each task. For instance, $H^2$ [34] performs element-wise parameter selection based on sensitivity measures, combined with a model search that utilizes continuous relaxation to explore the optimal architecture. Dynamic architecture methods either expand task-specific components as tasks increase, such as in [36], or use parallel sub-networks [37] to separate tasks. While these approaches effectively prevent interference between tasks, they come with significant trade-offs. Dynamic architectures, require increasing memory and thus computational complexity as tasks accumulate which can limit scalability in real-world applications. In contrast, fixed network architectures that isolate parameters for different tasks offer a more memory-efficient solution but sacrifice flexibility, as they restrict the model's ability to reuse parameters across tasks.

Recent work has advanced beyond traditional methods by leveraging self-supervised learning [31, 38, 39] and pre-trained networks [40–42] to develop robust representations, or by adjusting the optimization process [43, 44]. Additionally, some methods [13, 45] combine different strategies to create more effective solutions. For example, [45] applies regularization to task-specific latent distributions and replays both past inputs and distributions. $Co^2L$ integrates supervised contrastive learning for individual tasks with a self-supervised loss to transfer knowledge between old and new models.

There are also promising meta-learning approaches to continual learning, such as Meta Experience Replay (MER) [46], Online Meta-Learning (OML) [47], the Neuromodulated Meta-Learning Algorithm (ANML) [48], and La-MAML [49].

While the aforementioned continual learning methods are successful to some extent in mitigating forgetting, it is not clear whether regularization or isolation of parameters, distillation, or meta-learning will help in learning new unseen tasks. In fact, in regularization and parameter isolation approaches, parameters are identified as important by some form of evaluation on past tasks, without regard to whether these parameters will transfer to future tasks. Similarly, rehearsal-based approaches rely on some form of regularization or gradient alignment with respect to past task data to achieve good performance. While recent work [8, 9] considers features learned from new task data, they do not encourage learning of features that generalize to all tasks seen so far and are more likely to transfer. Similarly, recent meta-learning approaches such as La-MAML [49] use gradient-alignment heuristics to modulate the plasticity of parameters, but pay little attention to redundancy and the contribution of parameters to generalizability, and are computationally expensive compared to other continual

learning methods. Thus, there has been a general lack of attention to the transfer of continually learned knowledge to future tasks. To our interest, the recent approach named $Co^2L$ [31] questioned whether preserved past knowledge generalizes to future tasks and observed that contrastively learned representations [50, 51] transfer better and forget less, compared to learning based on the cross entropy loss.

Aiming for a continual learning approach that mitigates forgetting while learning representations that transfer well to unseen data, we were inspired to build on the contrastive learning framework [31, 51]. In contrastive learning, we work with an encoder mapping input images to vectors (*representations*), a projection head mapping representations to vectors (called *embeddings*) on which a contrastive loss is defined, and a decoder (linear transformation) mapping the extracted representations to class probabilities at inference time. We build our approach around $Co^2L$ [31], but importantly, we will selectively regularize the produced embeddings and network parameters based on how likely they are to transfer to future tasks. In doing so, we revisit assumptions about access to data at each point in time, and outline our inspiration from event models theorized to enable update of context representations in the brain.

**Task Boundaries and Event Models:** Events are based on how we understand the world around us. While the world appears to be a continuous stream of twists and turns, evidence suggests that we perceive it as discrete events at different spatiotemporal scales [52–54]. The brain has been theorized to operate and make sense of the world by updating and maintaining representations of the current situation, also known as *event models* [52–54]. Inspiring our work, event models are believed to be updated mainly at *event boundaries* [52–54]. These boundaries are thought to be detected by an increase in perceptual prediction error, i.e., when the brain's visual model makes predictions about the world that start to diverge from what is actually happening [52–54]. Interestingly, the said boundaries also exist in the field of continual learning at the moment the first batch of new task data arrives (or at any time when the model's prediction accuracy drops significantly). We will refer to these boundaries as *task boundaries*. While performing various types of computations during task boundaries is not new in continual learning, methods that perform such computations (e.g., EWC [2]) do not make use of all the information available at task boundaries. In the specific case of EWC [2], a regularization strength for each network parameter is computed using the data from the previous tasks, ignoring the first batch of data from the new task. To date, continual learning approaches have been focused on using previous task data and models to overcome forgetting. Assuming a stream of data where the data distribution changes, we can mark each time the model's performance on a batch of data drops as a task boundary and assign the data before this batch to the previous task. Consequently, the batch of data on which the model did not perform well will belong to a new task.

**Redundancy in Contrastive Learning:** Recent work suggests that most continual learning methods favor stability over plasticity, that is, they focus on not forgetting past tasks by preserving learned parameters and sacrificing flexibility to learn new knowledge [55]. Thus, it is advantageous to introduce less regularization into continual learning methods by retaining only parts of the learned network that are essential for performance on previous tasks *and* produce highly generalizable representations. Research on the properties of learned representations and the projection head of networks trained by contrastive loss has shown that over-parameterized (and sufficiently wide) neural networks learn embeddings with redundancy [56–58]. Specifically, the vector space in which the contrastive loss is defined is thought to suffer from a dimensional collapse problem [57, 58], i.e., the produced embeddings are in a lower-dimensional subspace of their nominal dimensionality. While this has been identified as an inefficiency in the normal supervised learning setting [57, 58], it provides an opportunity for continual learning: *regularization of DNN outputs can be defined only on parts of the embeddings instead of their entirety*. Similar to [56], we observe that a small subset of contrastively learned embeddings (i.e., a subset of output neurons combined) is able to replicate the performance of the entirety of embeddings on previous tasks. Moreover, we observe that different subsets of the embeddings of a DNN perform differently. By sampling random subsets of the embeddings produced by a DNN and evaluating them on previous and future tasks, we see that the variation in performance between subsets is higher on future tasks. These observations motivated us to define loss/regularization only on a small subset of the network's outputs, chosen so that it's likely to transfer to future tasks.

To select a highly generalizable subset, we propose to evaluate the network on the first batch of new task data (as a surrogate for unseen future data) during task boundaries. We introduce a novel procedure to identify the parts of the embeddings that perform best (a subset), and a novel loss

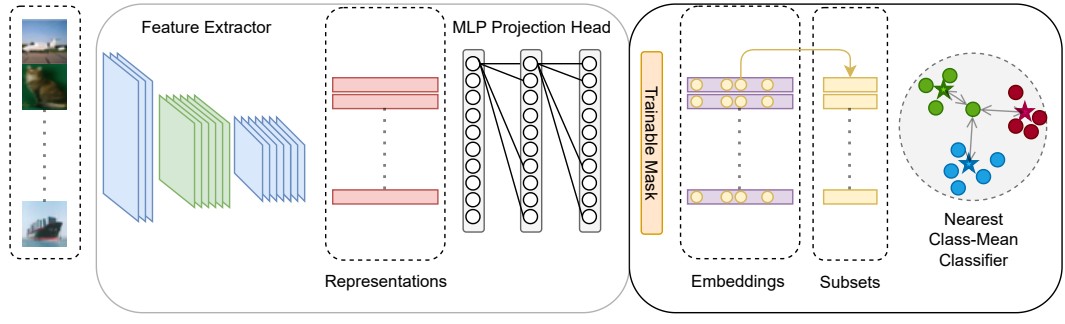

Figure 1: At task boundaries, the feature importance module is added on top of the embeddings to identify the salient subset. The mask marking the salient subset is trained based on a nearest class-mean classifier and regularized to be minimal (criterion 3).

to regularize this high-performing subset. We then introduce a novel extension of the excitation backprop [59] to measure the contribution of each network parameter in producing the identified subset, and a novel method to modulate the gradients based on this contribution. We will describe the details of our methods in the Methods section, followed by the experimental setup and results. In the ablation studies, we will justify our design choices and conclude with a discussion of the methods used and how they can be improved in the future.

## 2  Methods

We will use Co$^2$L [31] as our baseline and briefly review its methods. We will then build our proposed methods around it.

**Continual Learning Settings:** In continual learning, a model is trained on a sequence of tasks $\mathcal{T}_1, \mathcal{T}_2, ..., \mathcal{T}_n$. Each task is defined by its corresponding input and target datasets $(X_t, Y_t)$ which are drawn from a task-specific distribution $D_t$. Continual learning is mainly studied in three settings: task-incremental, class-incremental, and domain-incremental. In the task-incremental setting, the samples in each task are accompanied by a task identifier. As a result, during inference, a model can use the task identifier to drastically constrain target predictions. In the class- and domain-incremental settings, there is no knowledge of the task identifier at inference time, and the targets to be predicted can be any of the classes seen so far by the model. While the set of target classes in each task is disjoint in the task- and class-incremental settings, the set of classes remains the same in the domain-incremental setting (the $Y_t$ distribution stays the same while the $X_t$ distribution varies).

**Contrastive Learning and Co$^2$L Overview:** Supervised contrastive learning [51] generally involves a feature extractor mapping input samples to representations and a projection head mapping representations to embeddings. Formally, denoting a feature extractor parameterized by $\theta$ as $f_\theta$, representations by $\mathbf{r}$, projection head parameterized by $\psi$ as $g_\psi$, and embeddings by $\mathbf{e}$, supervised contrastive learning [51] and Co$^2$L [31] augment each input sample $x$ in the minibatch twice to get $\hat{x}_1$ and $\hat{x}_2$, known as views. Representations are generated by passing the views $\hat{x}_1$ and $\hat{x}_2$ to the feature extractor ($\mathbf{r}_1 = f_\theta(\hat{x}_1), \mathbf{r}_2 = f_\theta(\hat{x}_2)$). Embeddings are then created by passing the extracted representations to the projection head ($\mathbf{e}_1 = g_\psi(\mathbf{r}_1), \mathbf{e}_2 = g_\psi(\mathbf{r}_2)$). Both embeddings and representations are normalized to unit length ($|\mathbf{r}| = 1, |\mathbf{e}| = 1$). A contrastive loss is then defined on these embeddings and used to train the network. In the specific case of Co$^2$L [31], this loss is called the Asymmetric Supervised Contrastive Loss (Async SupCon) and is defined as:

$$\mathcal{L}_{\text{Async}}^{\text{SupCon}} = \sum_{i \in S} \frac{-1}{|\mathcal{P}_i|} \sum_{j \in \mathcal{P}_i} \log \left( \frac{\exp(\mathbf{e}_i \cdot \mathbf{e}_j / \tau)}{\sum_{k \neq i} \exp(\mathbf{e}_i \cdot \mathbf{e}_k / \tau)} \right)$$

where $S$ includes the index of views in the current task, $\mathcal{P}_i$ holds the index of views in the minibatch that belong to the same class as the $i$th view $\hat{x}_i$ except for $\hat{x}_i$ itself, $\tau$ is a temperature hyperparameter, and $\mathbf{e}_i$ is the embedding of the $i$th view. To facilitate comparison with previous work, we also use the Async SupCon loss to train the network.

To adapt supervised contrastive learning to solve a continual learning problem and to mitigate forgetting, Co$^2$L [31] uses an instance-wise relation distillation loss (IRD). IRD computes a similarity matrix by measuring the similarity of each view to other views in the minibatch (one row) for both the old model (a snapshot of the current model taken at the start of training on the current task and parameterized by $\omega$) and the model currently being trained (parameterized by $\theta$). The resulting two similarity matrices are then regularized to be similar to each other. Formally, the similarity of the views $\hat{x}_i$ and $\hat{x}_j$ is computed as follows:

$$R_{\theta,\eta_1}[i,j] = \mathrm{Sim}(\hat{x}_i, \hat{x}_j, \eta_1, \theta) = \frac{\exp(\mathbf{e}_i \cdot \mathbf{e}_j / \eta_1)}{\sum_{k \neq i}^{2N} \exp(\mathbf{e}_i \cdot \mathbf{e}_k / \eta_1)} \tag{1}$$

where $[i,j]$ denotes the element in the $i$th row and $j$th column of the pairwise similarity matrix, $\mathrm{Sim}$ is the similarity function, $\eta_1$ is a temperature hyperparameter, and $N$ is the number of samples in the minibatch. The IRD loss is then defined as:

$$\mathcal{L}_{\mathrm{IRD}} = \sum_{i=1}^{2N} \sum_{j=1}^{2N} -R_{\omega,\eta_2}[i,j] \cdot \log(R_{\theta,\eta_1}[i,j]) \tag{2}$$

We believe that this distillation loss is too restrictive and reduces the model's ability to learn new generalizable representations, since redundant parts of the embeddings are also regularized. We will modify this distillation loss so that it is only applied to a subset of embeddings. This subset will be identified by our novel *feature importance module* and regularized by the *selective distillation loss*. Similar to rehearsal-based continual learning approaches, we will use a small memory to store samples. The memory size will be similar to previous work for comparison and each class will be assigned an equal amount of memory. In addition, we extend the excitation backprop [59] framework to measure the contribution of individual network parameters in producing the identified salient subset as its salience. The proposed *gradient modulation* method will then use these salience values to decrease the gradients for salient network parameters. In the following sections, we will introduce the building blocks for these methods in more detail.

**Proposed Salient Subset Selection:** To improve the IRD loss [31], we try to identify a subset of the embeddings that satisfies the following criteria and refer to it as *salient*:

1. Transfers better to unseen data than other subsets (based on performance on unseen tasks),

2. Contains more information about past tasks compared to other subsets (based on performance on past tasks),

3. Is minimal, i.e., has no subset that performs as well on past and future tasks.

In the general continual learning formulation, criterion 1 and 2 cannot be evaluated for a subset, since we cannot store all the samples seen so far and future samples are yet to be seen. At a task boundary, however, we can use the samples stored in memory $\mathcal{M}$ as a surrogate for past tasks, and the first batch of new task data $\mathcal{B}_t$ (before the model is trained on it) as a surrogate for future tasks. We can create a dataset $\mathcal{D}_{\mathrm{SRS}}$ to use for finding the salient subset. This dataset can be created by combining $\mathcal{M}$ and $\mathcal{B}_t$ (the *combined* setting, $\mathcal{D}_{\mathrm{SRS}} = \mathcal{M} \cup \mathcal{B}_t$), or using memory samples only (the *onlypast* setting, $\mathcal{D}_{\mathrm{SRS}} = \mathcal{M}$), or using the first batch only (the *onlycurrent* setting, $\mathcal{D}_{\mathrm{SRS}} = \mathcal{B}_t$). These options will be evaluated in the ablation studies.

Identifying a salient subset of the embeddings is essentially a search problem. Here, for simplicity and speed, we adopt an approach similar to a previous work called Neural Similarity Learning [60], which represents which parts of the embeddings to select. Using the same notation as before, let $\mathbf{s}$ be a vector of the same size as $\mathbf{e}$, $\sigma$ denotes the sigmoid function, and $h_{\mathbf{s}}()$ be a Nearest Class Mean Classifier (NCMC) parameterized by $\mathbf{s}$ that takes in embeddings and assigns them to the class with the nearest mean embedding. Before training $h_{\mathbf{s}}()$, we need to compute the class means. Let $\mathcal{D}_c$ denote the samples in the dataset $\mathcal{D}_{\mathrm{SRS}}$ that belongs to class $c$, then the mean of class $c$ (denoted by $\mathbf{m}_c$) can be computed as:

$$\mathbf{m}_c = \frac{\sum_{(x,t) \in \mathcal{D}_c} g_\psi(f_\theta(x))}{|\mathcal{D}_c|} \tag{3}$$

where $x$ is a view of an input sample and $t$ is the class it belongs to.

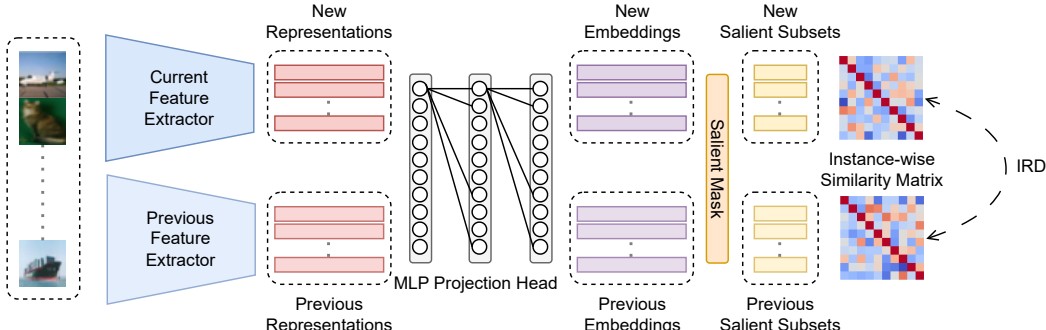

Figure 2: The instance-wise relation distillation loss (IRD) is applied only to a subset of embeddings deemed salient by the feature importance module.

To train $h_{\mathbf{s}}$ we will minimize the following loss function:

$$\ell_{\mathbf{s}}(\mathcal{D}_{\mathrm{SRS}}) = \frac{\sum_{(x,t)\in\mathcal{D}_{\mathrm{SRS}}} \frac{\mathbf{e}\odot\sigma(\mathbf{s})}{|\mathbf{e}\odot\sigma(\mathbf{s})|} \cdot \frac{\mathbf{m}_t\odot\sigma(\mathbf{s})}{|\mathbf{m}_t\odot\sigma(\mathbf{s})|}}{|\mathcal{D}_{\mathrm{SRS}}|} + \lambda|\mathbf{s}|_1 \tag{4}$$

where $\odot$ is the element-wise product. An $\ell_1$ norm loss (with $\lambda$ as a hyperparameter controlling the strength) is added to $\mathbf{s}$ to ensure that it marks a minimal subset (criterion 3). The size of this subset can vary depending on the redundancy of the embeddings. After training the NCMC for a number of randomly initialized mask vectors and selecting the best-performing mask, $\hat{\mathbf{s}} = \sigma(\mathbf{s})$ can be used to identify which parts of the embedding should be regularized. Compared to neural similarity learning [60], multiplying $\hat{\mathbf{s}}$ by the output of the encoder and the class means is similar to implementing a weighted dot product, weights that are used to mask parts of the encoder's output in our case.

**Proposed Selective Distillation:** *Selective Distillation* modifies the IRD loss so that it is only applied only to the salient subset of the generated embeddings. By selectively applying this distillation loss, we try to retain only parts of the embeddings that are salient, improving the model's flexibility in learning the new task, and promoting transfer and generalizability. Our proposed variant of IRD forms new embeddings $\hat{\mathbf{e}}$ by taking parts of the embeddings where the mask vector $\hat{\mathbf{s}}$ is above a threshold (here we simply choose 0.5):

$$\hat{\mathbf{e}} = \frac{\mathbf{e}[\hat{\mathbf{s}} \le 0.5]}{|\mathbf{e}[\hat{\mathbf{s}} \le 0.5]|} \tag{5}$$

The instance-wise similarity matrix (equation 1) is then computed using the new embeddings:

$$R_{\theta,\eta_1}[i,j] = \mathrm{Sim}(\hat{x}_i, \hat{x}_j, \eta_1, \theta) = \frac{\exp(\hat{\mathbf{e}}_i \cdot \hat{\mathbf{e}}_j / \eta_1)}{\sum_{k\neq i}^{2N} \exp(\hat{\mathbf{e}}_i \cdot \hat{\mathbf{e}}_k / \eta_1)} \tag{6}$$

The IRD loss (equation 2) is then calculated using the new instance-wise similarity matrices.

**Salient Parameter Selection:** After identifying the salient subset, the computed salience can be passed down using a novel extension of excitation backprop (EB) [59]. Normally, EB is a method that attributes the *activation* of a model's output neurons to its *input*. Our goal, however, is different: we want to attribute the *performance* of salient neurons in the output layer to individual network *parameters* given a batch of data samples.

Assuming a simple neuron computes $a_i^{l+1} = \phi(\sum_j w_{j,i}^l a_j^l)$ where $a_i^{l+1}$ is the activation value of the $i$th neuron in the $(l+1)$th layer, $w_{j,i}^l$ is the weight connecting the $j$th neuron in the $l$th layer to the $i$th neuron in the $(l+1)$th layer, and $\phi$ is a non-linear activation function. EB defines the salience of a neuron's activation as its *winning probability* $P(a)$. To compute salience, it uses the marginal winning probability (MWP) of a neuron, given neurons in the upper layer:

$$P(a_j) = \sum_{a_i\in\mathcal{P}_j} P(a_j|a_i)P(a_i) \tag{7}$$

$\mathcal{P}_j$ denotes the neurons in the layer above (closer to the output) $a_j$. Under certain assumptions (see [59], which holds when the ReLU activation function is being used), the MWP for a neuron $a_j^l$ can be computed based on the salience of neurons $a_i^{l+1}$ in the upper layer:

$$P(a_j^l | a_i^{l+1}) = \begin{cases} Z_i a_j^l w_{j,i}^l & \text{if } w_{j,i}^l \geq 0, \\ 0 & \text{otherwise.} \end{cases} \tag{8}$$

Where $Z_i$ is a normalization factor and is equal to $\frac{1}{\sum_{j, w_{j,i}^l \geq 0} a_j^l w_{j,i}^l}$. Using the MWPs computed from Eq. 8, the salience of each neuron can be computed in top-down order based on Eq. 7.

To attribute the salience of output neurons to the model's *parameters*, similar to [6] we first use EB to compute the salience for activation maps in each layer. Next, similar to Oja's rule [61], the salience of each network weight can be computed using the salience of its two ends:

$$\gamma(w_{i,j}^l) = \sqrt{P(a_i^l) P(a_j^{l+1})} \tag{9}$$

where $\gamma$ represents salience. The output of this *salient parameter selection* process is essentially a salience value for each network parameter. These salience values will be used in the next step to modulate gradients.

**Gradient Modulation:** Inspired by neuromodulation processes in the brain, where the plasticity of neurons can change depending on the task at hand [62], we try to limit the change of network weights that are considered salient. In the domain of neural networks, this means modifying gradients so that the more salient a network weight is, the less the gradient is modified. To achieve this, we modify the gradients as follows:

$$d_w = d_w \times (1 - \min(1, \gamma(w))) \tag{10}$$

where $d_w$ is the gradient with respect to parameter $w$. This process aims to guide the network during training by shifting its focus on learning the task at hand using parameters that did not contribute to the performance of the salient subset.

## 3 Results

**Evaluating Random Subsets:** We test the hypothesis that it is beneficial to use the first batch of new task data at task boundaries. Specifically, we want to see if subsets of network-generated embeddings have the same discrimination power with respect to past versus future tasks. At each task boundary, we extract 10 random neurons from the network-generated embedding to form a subset. Using only the selected subset, we first train a linear classifier to discriminate between classes in the entirety of past tasks' data, and then train another linear classifier using the same subset of neurons to discriminate between classes in the entirety of unseen tasks' data. We repeat this process 100 times and record the accuracy of the selected subset on past and unseen task data. We then compute the mean and variance of the subset accuracy across these 100 subsets. We observed a higher variance when evaluating on unseen tasks (see appendix A.1 for details), suggesting that the generalizability of subsets varies more than their captured knowledge of past tasks.

**Proposed Method Results:** To allow comparison with previous results [31], we conduct experiments in the task-incremental, class-incremental, and domain-incremental settings on the CIFAR-10 [63], TinyImageNet [64], and R-MNIST datasets [17] (see A.2 for experimental setup details). We compare our results with rehearsal-based continual learning methods, including Co$^2$L [31], ER [46], iCaRL [15], GEM [17], A-GEM [65], FDR [66], GSS [16], HAL [67], DER [68], and DER++ [68]. A low (200 samples) and a high (500 samples) memory setting were considered. Results are the average test-set classification accuracy on all seen classes at the end of training.

We compare the results of our proposed methods with previous work in table 1. We use SD to refer to our selective distillation method, GM to refer to our gradient modulation method when the IRD loss is applied as in [31] rather than selectively as in SD, and SD + GM to refer to the simultaneous use of gradient modulation and selective distillation. SD is superior to baselines and state-of-the-art for both task and class-incremental settings on the SplitCIFAR10 and SplitTinyImageNet datasets. It also outperforms previous work on the domain-incremental setting on the R-MNIST dataset when using small memory. GM and SD+GM also improved the state-of-the-art on the SplitCIFAR10 dataset, but did not outperform SD. A discussion of GM is provided in appendix A.3. These results show that SD

Table 1: Comparison of our proposed methods with published methods. Proposed methods were run with the onlycurrent setting of salient subset selection and their accuracy was obtained by averaging across 5 independent trials. The highest accuracy is in bold. '-' denotes settings where evaluation was not possible due to incompatibility or intractable training processes. Previous results listed are based on [31]. Data are presented as mean (SD).

| Memory Size | Dataset | SplitCIFAR10 | | SplitTinyImageNet | | R-MNIST |
| --- | --- | --- | --- | --- | --- | --- |
| | Scenario | Class-IL | Task-IL | Class-IL | Task-IL | Domain-IL |
| 200 | ER | 44.79 (1.86) | 91.19 (0.94) | 8.49 (0.16) | 38.17 (2.00) | 93.53 (1.15) |
| | GEM | 25.54 (0.76) | 90.44 (0.94) | - | - | 89.86 (1.23) |
| | A-GEM | 20.04 (0.34) | 83.88 (1.49) | 8.07 (0.08) | 22.77 (0.03) | 89.03 (2.76) |
| | iCaRL | 49.02 (3.20) | 88.99 (2.13) | 7.53 (0.79) | 28.19 (1.47) | - |
| | FDR | 30.91 (2.74) | 91.01 (0.68) | 8.70 (0.19) | 40.36 (0.68) | 93.71 (1.51) |
| | GSS | 39.07 (5.59) | 88.80 (2.89) | - | - | 87.10 (7.23) |
| | HAL | 32.36 (2.70) | 82.51 (3.20) | - | - | 89.40 (2.50) |
| | DER | 61.93 (1.79) | 91.40 (0.92) | 11.87 (0.78) | 40.22 (0.67) | 96.43 (0.59) |
| | DER++ | 64.88 (1.17) | 91.92 (0.60) | 10.96 (1.17) | 40.87 (1.16) | 95.98 (1.06) |
| | Co$^2$L | 65.57 (1.37) | 93.43 (0.78) | 13.88 (0.40) | 42.37 (0.74) | 97.90 (1.92) |
| | **SD (ours)** | **73.72 (0.52)** | **96.10 (0.09)** | **16.02 (0.39)** | **44.07 (0.66)** | **98.80 (0.26)** |
| | **GM (ours)** | 71.30 (1.15) | 95.84 (0.25) | 12.46 (0.43) | 38.33 (0.90) | 97.29 (0.59) |
| | **SD + GM (ours)** | 70.64 (0.98) | 95.28 (0.46) | 12.93 (0.55) | 38.47 (0.68) | 96.68 (0.55) |
| 500 | ER | 57.75 (0.27) | 93.61 (0.27) | 9.99 (0.29) | 48.64 (0.46) | 94.89 (0.95) |
| | GEM | 26.20 (1.26) | 92.16 (0.64) | - | - | 92.55 (0.85) |
| | A-GEM | 22.67 (0.57) | 89.48 (1.45) | 8.06 (0.04) | 25.33 (0.49) | 89.04 (7.01) |
| | iCaRL | 47.55 (3.95) | 88.22 (2.62) | 9.38 (1.53) | 31.55 (3.27) | - |
| | FDR | 28.71 (3.23) | 93.29 (0.59) | 10.54 (0.21) | 49.88 (0.71) | 95.48 (0.68) |
| | GSS | 49.73 (4.78) | 91.02 (1.57) | - | - | 89.38 (3.12) |
| | HAL | 41.79 (4.46) | 84.54 (2.36) | - | - | 92.35 (0.81) |
| | DER | 70.51 (1.67) | 93.40 (0.39) | 17.75 (1.14) | 51.78 (0.88) | 97.57 (1.47) |
| | DER++ | 72.70 (1.36) | 93.88 (0.50) | 19.38 (1.41) | 51.91 (0.68) | 97.54 (0.43) |
| | Co$^2$L | 74.26 (0.77) | 95.90 (0.26) | 20.12 (0.42) | **53.04 (0.69)** | **98.65 (0.31)** |
| | **SD (ours)** | **76.49 (0.63)** | **96.39 (0.20)** | **21.49 (0.50)** | 52.69 (0.45) | 98.43 (0.38) |
| | **GM (ours)** | 74.63 (0.95) | 96.15 (0.14) | 17.54 (0.44) | 48.21 (0.54) | 97.17 (0.50) |
| | **SD + GM (ours)** | 73.82 (0.42) | 95.67 (0.14) | 19.01 (0.31) | 48.06 (0.71) | 96.49 (1.15) |

can successfully mitigate forgetting while freeing up the remaining parts of the model to learn new tasks. In the next section, we will analyze the choice of selecting the salient subset based only on the new batch of data, rather than on memory samples or combined. We will also go over the effect of the embedding size for our method (SD) as it depends on the redundant units in the output of the projection head (embeddings).

## 4 Ablation Studies

**Identifying the Salient Subset of Embeddings, onlycurrent, onlypast, or combined:** Although the proposed methods outperformed published methods, it was unclear which parts of our approach contributed to the performance gain. In salient subset selection, three settings were used to generate $\mathcal{D}_{\text{SRS}}$. The salient subset was then chosen based on the classification performance of subsets on $\mathcal{D}_{\text{SRS}}$. Initially, we hypothesized that including the first batch of new task data would help the salient subset selection identify parts of the embeddings that not only perform well on previous tasks but also generalize well to unseen tasks. To examine this hypothesis, we conducted experiments (Results in Table 2) on the SplitTinyImageNet and SplitCIFAR10 datasets in these three settings using the selective distillation method.

For the SplitCIFAR10 dataset, the onlycurrent setting where $\mathcal{D}_{\text{SRS}} = \mathcal{B}_t$ outperformed the onlypast and combined settings. The lower performance of the combined setting compared to the onlypast setting can be explained by the low number of classes in the CIFAR10 dataset. Added memory samples in the $\mathcal{D}_{\text{SRS}}$ dataset may be misleading as a significant portion of memory samples will belong to the task the model was just trained on. The performance of various parts of embeddings on the previous task may be less informative as it measures neither resilience to forgetting nor generalizability.

Table 2: Comparison of SD performance for different settings of the salient subset selection process. The onlycurrent setting uses the first batch of the new task, onlypast uses samples in the memory, and combined uses both for identification of the salient subset in embeddings. Adding the first batch of new task data improves SD performance in virtually all scenarios and datasets. Five independent experiments were conducted for each case to report the mean and variance. A memory buffer of 500 samples was used in all experiments.

| Dataset | SplitCIFAR10 | | SplitTinyImageNet | |
|---|---|---|---|---|
| Setting | Class-IL | Task-IL | Class-IL | Task-IL |
| onlypast | 75.33 (0.53) | 96.28 (0.15) | 21.42 (0.25) | 52.64 (0.55) |
| combined | 75.20 (0.88) | 96.29 (0.17) | 22.07 (0.37) | 52.78 (0.35) |
| onlycurrent | 76.49 (0.63) | 96.39 (0.20) | 21.49 (0.50) | 52.69 (0.45) |

Experimenting on the SplitTinyImageNet dataset, we observed that both the onlycurrent and combined settings outperformed the onlypast setting. It is worth emphasizing that the onlycurrent setting outperformed the onlypast setting on both datasets and continual learning scenarios, suggesting that using a batch of new task data may be useful for identifying the salient subset. Also note that all accuracies listed in Table 2 were higher than previous state-of-the-art results [31], demonstrating that while changing the default continual learning protocol to use the first batch of new task data may improve model performance to some extent, the main performance gains were results of the selective distillation (SD) method itself.

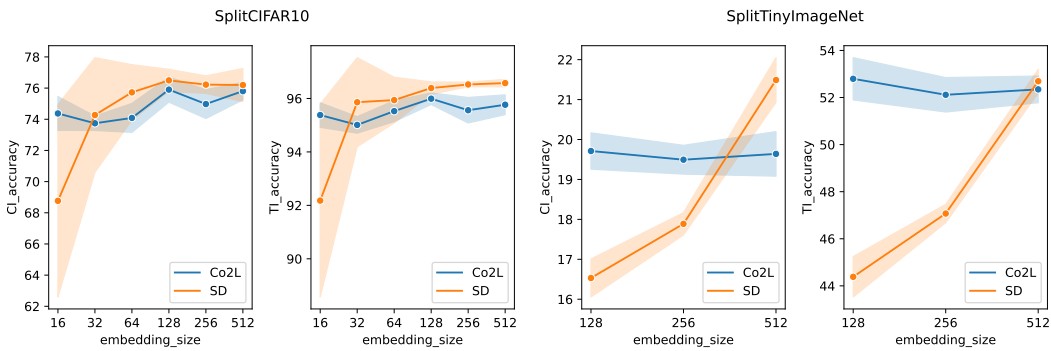

Figure 3: Comparing SD (ours) and Co$^2$L [31] using different embeddings sizes on the SplitCIFAR10 and SplitImagenNet datasets. The memory size is the same (500) for both methods. Shading depicts standard deviation. Increasing embedding size and redundancy benefits SD on both datasets.

**The Effect of the Embedding Size:** In our first experiments, we noticed that SD outperformed Co$^2$L [31] on all datasets except for SplitTinyImageNet. Our hypothesis was that SD relied on redundancy in the embeddings and when the generated embeddings were dense, it was reasonable to apply the IRD loss on entire embeddings rather than a subset. Moreover, since in contrastive learning the projection head is discarded after training and is generally small (MLP, 512 hidden units, 128 output units in Co$^2$L), increasing embedding size to induce redundancy comes with virtually no computational cost, especially at inference time. To test our hypothesis, we compared SD to Co$^2$L [31] with different embedding sizes on the SplitCIFAR10 and SplitTinyImageNet datasets. For SplitCIFAR10, the embeddings seemed to be dense when the embedding size was around 16 and started to involve some redundancy starting from 32 units in the output (figure 3 left). As we increased the embedding size starting with 32 units, we noticed that SD consistently outperforms Co$^2$L [31].

When testing our hypothesis on the SplitTinyImageNet dataset (which is generally more difficult to solve with 200 classes), we noticed that embeddings appeared to be dense until an embedding size of 256 and SD was unable to outperform Co$^2$L [31]. However, with an embedding size of 512, redundancy began to materialize in embeddings and SD achieved higher task- and class-incremental accuracy (figure 3 right). We did not increase the embedding size further as it would have gotten larger than the hidden layer's size and could have caused complications unrelated to this ablation

study. Overall, these results showed that as the embedding size grows larger, SD can leverage the increased redundancy and improve continual learning performance in both task and class-incremental settings.

## 5 Conclusion and Future Work

Inspired by event models, we proposed a different way of looking at the continual learning setting, focusing on task boundaries. We hypothesized that the first batch of new task data could be used to identify parts of the neural network that enable generalization to unseen tasks. Observing the redundancy-inducing effects of the contrastive loss on embeddings, we first introduced a salient subset selection process to identify a subset that performs similarly to the full set of embeddings. Secondly, we presented a selective distillation method that regularizes only the salient parts of the embeddings. Thirdly, we introduced an attribution method that assigned salience to network parameters based on their contribution to the computation of the salient subset. Fourthly, we proposed a gradient modulation method that modified gradients according to the salience of parameters. Our methods did not increase parameters linearly with the number of tasks, nor did they assume that additional memory was available in the form of a second snapshot of the model or more samples in memory. Moreover, in alignment with our hypothesis, the selective distillation method was able to leverage redundancy in the embeddings and demonstrated superior performance compared to previous work. Further studies on the properties of projection heads in representation learning can open new avenues for methods like selective distillation to better separate task-specific knowledge. Additionally, modifications to our gradient modulation technique present a promising direction. An avenue for improvement involves adjusting gradient modulation to induce redundancy in a layer-wise manner, aligning the degree of parameter regularization with each layer's role in learning new tasks. Such adjustments would enhance knowledge consolidation across the network while preserving plasticity, thereby reducing forgetting and ultimately improving the overall performance of continual learning.

## Acknowledgments and Disclosure of Funding

R.M. acknowledges support from the Vector Scholarship in Artificial Intelligence, provided through the Vector Institute, and Western University via the Western Graduate Research Program (WGRS). J.M. acknowledges support from a BrainsCAN Postdoctoral Fellowship Award through the Canada First Research Excellence Fund (CFREF). Y.M. acknowledges support from BrainsCAN at Western University through the Canada First Research Excellence Fund (CFREF) and Natural Sciences and Engineering Research Council of Canada Discovery Grant.

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

# A    Appendix / supplemental material

## A.1    Variance in Subset Accuracy

For the purposes of this analysis only, we will deviate from the standard continual learning protocol. Note that our main proposed methods follow the standard continual learning protocol. Training a linear classifier on previous and upcoming task data using 100 random embedding subsets of size 10 we observed that the performance of different random subsets has a noticeable variation on both past and future tasks, but the variance is generally higher on future tasks (table 3). Taking accuracy on future tasks as an indicator for generalizability of a subset, this also shows that not all subsets are equal in terms of how generalizable they are, and thus regularizing parts of the embeddings that do not transfer well is limiting the network's ability to learn new tasks. These results support our decision to include the first batch of new task data to identify the salient subset.

Table 3: Mean (std) of subset accuracy on previous and upcoming tasks are evaluated at each task boundary. The results are computed based on three independent trials on the SplitCIFAR10 dataset. The standard deviation for these results is calculated based on trials.

|  | Task 1 | Task 2 | Task 3 | Task 4 |
| --- | --- | --- | --- | --- |
| Mean subset accuracy on past tasks | 99.18 (0.06) | 74.19 (0.34) | 63.09 (4.35) | 54.08 (0.97) |
| Mean subset accuracy on future tasks | 30.50 (0.53) | 38.65 (0.62) | 60.56 (0.40) | 76.81 (0.69) |
| Std of subset accuracy on past tasks | 0.23 (0.01) | **3.75 (0.10)** | 2.70 (0.40) | 2.92 (0.09) |
| Std of subset accuracy on future tasks | **2.10 (0.03)** | 2.54 (0.08) | **3.82 (0.42)** | **4.99 (0.45)** |

## A.2    Experimental Details

We conduct experiments in three common continual learning scenarios: Task-Incremental (Task-IL), Class-Incremental (Class-IL), and Domain-Incremental (Domain-IL). For class and task-incremental settings, CIFAR-10 [63] and TinyImageNet [64] datasets were used, while for the domain-incremental setting, we used Rotational MNIST (R-MNIST) [17]. CIFAR-10 and TinyImageNet will be divided across classes into 5 and 10 sub-datasets to create SplitCIFAR10 and SplitTinyImageNet respectively. Each task will then be to solve an image classification task on 2 classes for SplitCIFAR10 and 20 classes for SplitTinyImageNet. The order of classes is the same across experiments. The R-MNIST dataset will consist of 20 tasks, where for each task the MNIST [69] dataset is rotated using a random degree in the range of $[0, \pi)$ (uniformly sampled). Similar to [31], when training on R-MNIST, the same digits rotated by a random degree will be treated as different classes in the Async SupCon loss. The implementation for this work is based on the implementation of [31]. Unless otherwise stated, all choices of optimizer, architecture, and hyperparameters were kept the same.

For training on SplitCIFAR10 and SplitTinyImageNet, we use the ResNet-18 [70] architecture while for R-MNIST, the same smaller architecture as in [31] is employed for comparison. A two-layer linear network is used for the projection head. Importantly, we increase the embedding size (output of projection head) for the SplitTinyImageNet dataset. We have explained this design choice in the ablation study 4. Evaluation is according to contrastive learning framework [31, 51] which trains a classifier on top of the encoder using last task samples and samples in the memory (as if the classifier was trained immediately after learning a task, according to samples available at the time). We used two Nvidia RTX 3090 GPUs for training and evaluation. Training time for the SplitCIFAR10 dataset was about 2 hours, for SplitTinyImageNet around 8 hours, and for R-MNIST about an hour.

## A.3    Discussion on Gradient Modulation

Neuromodulation-inspired mechanisms have enabled continual adaptation in a wide range of tasks, including navigation [71, 72], language modeling [73], and image classification [74]. Similarly, our approach used saliency information to modify the gradients for parameters that are identified as salient.

Our main motivation in designing gradient modulation was to find a measure of importance based on a parameter's contribution to formation of embeddings that transfer well. In addition to this main motivation, we had three goals in designing GM the way we did. First, we wanted it to be biologically plausible similar to EWC [2]. Second, unlike EWC [2] and AFEK [8] we wanted it to be aware of the

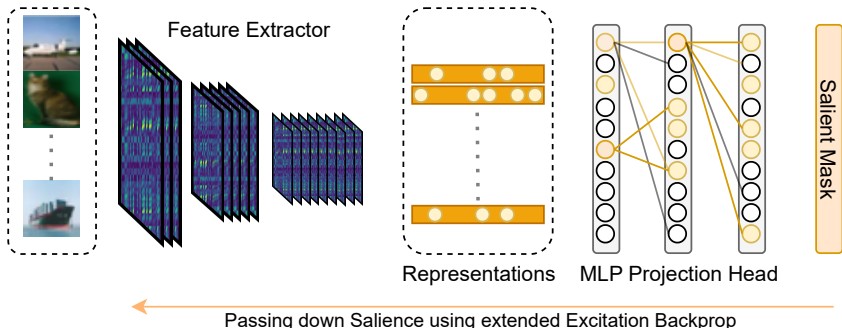

Figure 4: Using the mask vector $\hat{\mathbf{s}}$ to pass down salience to each network weight using our extended version of excitation backprop [59]. The computed parameter importance is then used to modulate gradients.

structure of the neural network and regularize a parameter only if it contributes to the activation of an important parameter in the upper layer. Third, observing that methods like EWC [2] favor stability over plasticity [55] and are unable to learn new tasks effectively, we aimed to design a method that limited regularization to as few parameters as possible while leaving most of the network free to learn new generalizable knowledge. We believe that this last goal was not met as well as we aimed for, mainly because of the interconnectedness of neural networks. Specifically, when an important subset is identified within the embedding layer, this subset is minimal by design (see the Proposed Salient Subset Selection section2). However, when excitation backprop is applied top-down iteratively to find parameters that contribute to the formation of salient parameters in the upper layer, the number of parameters identified as salient grows larger at each layer. As EB approaches the input layer, almost all parameters contribute to the salient subset in the embedding layer in one way or another and are identified as salient. Thus, while the method has more plasticity compared to previous work in the layers near the output, it suffers from too much stability in layers near the input. To mitigate this, GM should pick gradients to modulate more sparsely. To achieve this, one may need to effectively separate knowledge and features learned within a neural network. As the changes needed become more complex, we believe that it is better to examine them in future work.

Although our gradient modulation surpassed state-of-the-art performance on the SplitCIFAR10 dataset with or without selective distillation, it did not perform better than SD. While it may seem that GM is not a promising technique for continual learning, it is worth noting that our extension of the excitation backprop [59] provides useful saliency and attribution information. It can compute for each network parameter a salience value that describes its contribution in forming a specific subset in the embeddings. We used the parameter salience information produced by this framework to modulate gradients to encourage learning new tasks using parts of the network that did not seem to contribute to salient features in the embeddings. However, the parameter salience information can be used in many different ways, e.g., identifying parameters to regularize and preserve, finding subnetworks that are capable of performing similarly to the whole network, or finding the least salient parameters. To our knowledge, this method is the only variant of excitation backprop [59] that can be used in networks where the loss function is defined on representations or embeddings (representation learning). It is also worth noting that this method can assign salience based on the performance of the generated embeddings and representations, not just the activations of certain neurons. Overall, we believe that GM is a multipurpose tool with use cases that go beyond continual learning.

### A.4 Discussion on Memory and Compute Usage

Following the general continual learning desiderata [75] we focused on using a fixed-capacity model. As a result, we did not include model-growing approaches such as Progressive Neural Networks [76] and TAMiL [77] in our reported results. We also did not consider multiple memory approaches where the memory usage goes further than a copy of the main model and a memory of samples. These approaches include a recent promising work called CLS-ER [18] where two exponentially averaged copies of the model are maintained. Although our approach outperforms CLS-ER on the SplitCIFAR10 dataset, we believe methods with two memory systems should be compared with each

other, and single memory systems should be compared with one another for a fair comparison. A copy of the ResNet-18 architecture is typically 40 MB in size, while each image in the TinyImageDataset is about 3 KB. The low memory setting assumes access to memory is so limited that only 200 samples (one per class) can be stored. The addition of a copy of a ResNet-18 model is similar to adding more than 10,000 samples to this memory and thus gives a significant advantage compared to a method that employs one copy only.

Furthermore, empirical results in recent work [18, 77, 78] suggest that using an exponentially averaged model over the trajectory of learning is more robust in mitigating forgetting compared to using a static snapshot of the model from a single point in time. However, to emphasize the robustness of our methods, we decided to test them in a standalone manner and did not use this technique. We leave it to future work to combine our methods with CLS-ER [18] or ESMER [78] and study the effects.

## A.5 Computational Complexity of the Gradient Modulation Method

The proposed GM is implemented based on excitation backprop [59], computing importance for weights in addition to activations. To compute the importance of activations, first, a forward pass takes the inputs to the network and computes layer activations. EB [59] then performs a backward pass, computing the importance of activations from top to bottom. In each layer, EB computes raw importance values for the lower layer and then performs a mini-forward pass to normalize these importance values. We perform an additional mini-forward and a mini-backward pass to attribute the importance of activations to layer weights. This is independent of the layer type and works based on Pytorch's autograd functionality. As a result, GM performs two forward and backward passes to compute the salience of the network parameters. Note that this computation is performed only once during task boundaries and does not occur during training on a task.

