# OpenReview forum: "Look-Ahead Selective Plasticity for Continual Learning of Visual Tasks"
_NeurIPS.cc/2024/Workshop/UniReps — UniReps_

### Official Review · Reviewer_DJoP · 2024-10-04
**Interesting approach to continual learning based on contrastive learning**

**Rating:** 7
**Confidence:** 4

**Review:**

This submission points out that most continual learning methods focus on avoiding catastrophic forgetting of past tasks and pay little or no attention to the generalizability of a model for future tasks. The authors present a replay-based continual learning approach that considers both goals, adopting  the contrastive learning framework. To adapt supervised contrastive learning to solve a continual learning problem and to mitigate forgetting, Co2L [31] uses an instance-wise relation distillation loss (IRD). This submission modifies the IRD loss so that it applies only to a sub set of the embeddings.

The proposed method consists of two parts:
- Salient subset selection
Using stored samples at the task boundary as a surrogate for past tasks, and the first batch of new task data as a surrogate for future tasks, a subset of embeddings is identified
- Salient parameter selection
 The computed salience of embeddings dimensions is passed down using an extension of excitation backprop.

The submission presents a fairly original approach to continual learning, explicitly considering the generalizability to future tasks. The authors  employ contrastive learning as a tool to ensure generalizability, which makes sense, given its superior ability to learn representations that generalize to downstream tasks. The presentation of the method is clear. Related work is discussed comprehensively.

In their experiments, the authors compare their proposed method against 10 baselines on three continual learning benchmark datasets for two different memory sizes. The experimental design is thorough and fair. The experimental results show that the proposed method outperforms the closely related method Co2L in some but not all of the scenarios.

The authors do not discuss directions for future research, which is somewhat unfortunate for a workshop paper.

Overall, an interesting and solid paper, which promises to stimulate discussions at the workshop.

---

### Official Review · Reviewer_2fTp · 2024-10-06
**Accept but needs rewritting and a few changes**

**Rating:** 5
**Confidence:** 5

**Review:**

This paper addresses the challenge of catastrophic forgetting in continual learning by introducing a novel approach focused on preserving generalizable knowledge while promoting plasticity for new tasks. Their pipeline looks like this:

Identify important parts of the output (Salient Subset Selection)
Preserve these important parts (Selective Distillation)
Identify which parameters are responsible for these important parts (Salient Parameter Selection)
Adjust learning based on parameter importance (Gradient Modulation)

Contrastive Learning: While the authors' main contributions are in selectively preserving and updating knowledge (through salient subset selection and selective distillation), the underlying learning framework is based on contrastive learning principles.

Reasonable experiments were conducted which gave decent results.

Some questions and suggestions I had:

1. FIM/EWC/AFEC already solves this problem of Salient Parameter Selection. Why not replace your saliency/gradient modulation stuff with that and see what happens? Why should someone use your method vs FIM/EWC/AFEC? AFEC was cited but I didn't see any computational cost or accuracy comparison against it.

2. Section 2: All the sub-sections seem to be doing related tasks. It will be easier if you cut down on the text and add a figure that shows an easy-to-read pipeline instead.

---

### Official Review · Reviewer_E75H · 2024-10-07
**Solid work demonstrating step-improvement of baseline continual learning work based on finding salient representation regions at the task boundaries.**

**Rating:** 7
**Confidence:** 3

**Review:**

## 1. Summary

The work extends the contrastive continual learning method Co2L by determining salient and non-salient representation regions based on task boundaries. Additionally, the work explores a novel technique to not only identify salient activation representations but also apply gradients to weights modulated according to their saliency. While the method based on salient activations ("Selective Distillation") shows promising, state-of-the-art results with improvements over the baseline Co2L method, the weight saliency approach does not seem to further improve the results.

## 2. Strengths and Weaknesses

The work demonstrates improvements on the baseline Co2L method with a clear, elegant idea of targeting modulations based on saliency scores computed at the task boundaries. The results show some improvements over the baseline approach, with ablations testing the composition of the memory buffer. Further ablations would be appreciated to test the key components in the method that generates the saliency scores.

The results of the weight saliency approach should be analysed in more detail, given it is used as part of the novelty claim for the paper, but does not show an improvement. Why is it not working as expected?

The paper is clearly written, and the experiments are well-presented.

Given the interest in continual learning, this work demonstrates a noticeable improvement to an existing baseline, both conceptually and experimentally, with relatively minor changes to the original equations (which I see as a plus).

## 3. Questions

- What is the effect of changes to the saliency score generation method (beyond the tested data composition)?
- Given that the results show that it is not imperative to include the first batch of new tasks, could this method also be applied in a scenario with continually changing tasks and/or domains (without clear boundaries)?

## 5. Ethical Concerns

None

## 6. Soundness

3 – Good

## 7. Presentation

4 – Excellent

## 8. Contribution

3 – Good

---

### Decision · Program_Chairs · 2024-10-10

**Decision:**

Accept

**Comment:**

In light of the positive reviewers' feedback and relevancy of the submission, we are pleased to accept this paper for presentation at UniReps 2024. We kindly ask the authors to incorporate the reviewers' suggestions and feedback in the final camera-ready version of the manuscript.